# Regulatory T Cells in Pancreatic Cancer: Of Mice and Men

**DOI:** 10.3390/cancers14194582

**Published:** 2022-09-21

**Authors:** Carmen Mota Reyes, Elke Demir, Kaan Çifcibaşı, Rouzanna Istvanffy, Helmut Friess, Ihsan Ekin Demir

**Affiliations:** 1Department of Surgery, Klinikum Rechts der Isar, School of Medicine, Technical University of Munich, 81675 Munich, Germany; 2German Cancer Consortium (DKTK), Partner Site Munich, 81675 Munich, Germany; 3CRC 1321 Modelling and Targeting Pancreatic Cancer, Klinikum Rechts der Isar, School of Medicine, Technical University of Munich, 81675 Munich, Germany; 4HPB-Unit, Department of General Surgery, School of Medicine, Acibadem Mehmet Ali Aydinlar University, Istanbul 34752, Turkey

**Keywords:** regulatory T cells, pancreatic cancer, neoadjuvant therapy, myeloid derived suppressor cells, immunotherapy

## Abstract

**Simple Summary:**

Regulatory T cells (Treg) are a major immunosuppressive cell subset in the pancreatic tumor microenvironment. Tregs influence tumor growth by acting either directly on cancer cells or via the inhibition of effector immune cells. Treg cells form a partially redundant network with other immunosuppressive cells such as myeloid-derived suppressor cells (MDSC) that confer robustness to tumor immunosuppression and resistance to immunotherapy. The results obtained in preclinical studies, whereupon Treg depletion, MDSCs concomitantly decreased in early tumors whereas an inverse association was seen in advanced PCa, urge a comprehensive analysis of the immunosuppressive profile of PCa throughout tumorigenesis. One relevant context to analyse these compensatory mechanisms may be patients with locally advanced PCa undergoing neoadjuvant therapy (neoTx). In order to understand these dynamics and to uncover stage-specific actional strategies involving Tregs, pre-clinical models that allow the administration of neoTx to different stages of PCa may be a very useful platform.

**Abstract:**

Regulatory T cells (Treg) are one of the major immunosuppressive cell subsets in the pancreatic tumor microenvironment. Tregs influence tumor growth by acting either directly on cancer cells or via the inhibition of effector immune cells. Treg cells mechanisms form a partially redundant network with other immunosuppressive cells such as myeloid-derived suppressor cells (MDSC) that confer robustness to tumor immunosuppression and resistance to immunotherapy. The results obtained in preclinical studies where after Treg depletion, MDSCs concomitantly decreased in early tumors whereas an inverse association was seen in advanced PCa, urge a comprehensive analysis of the immunosuppressive profile of PCa throughout tumorigenesis. One relevant context to analyse these complex compensatory mechanisms may be the tumors of patients who underwent neoTx. Here, we observed a parallel decrease in the numbers of both intratumoral Tregs and MDSC after neoTx even in locally advanced PCa. NeoTx also led to decreased amounts of αSMA^+^ myofibroblastic cancer-associated fibroblasts (myCAF) and increased proportions of CD8^+^ cytotoxic T lymphocytes in the tumor. In order to understand these dynamics and to uncover stage-specific actional strategies involving Tregs, pre-clinical models that allow the administration of neoTx to different stages of PCa may be a very useful platform.

## 1. PCa

PCa is one of the deadliest human neoplasms and is projected to become the second leading cause of cancer-related death worldwide by 2030 [1]. Its poor prognosis is partially due to the fact that most patients have metastatic disease and an overwhelming resistance to most cancer therapies, including current modalities of immune checkpoint blockade [1,2]. The therapeutic failure in PCa results from a low level of immunogenicity of cancer cells, the tumor’s robust immunosuppressive machinery, or both [1]. In PCa, the enrichment of suppressive immune cells was found to occur already at the precursor stage, i.e., around pancreatic intraepithelial neoplasia (PanIN) lesions and their prevalence increases throughout tumor progression [1,3]. The major contributors to the pro-tumorigenic features of the pancreatic tumor microenvironment include a strongly fibrotic stroma and an accumulation of immunosuppressive cell populations such as regulatory T cells (Tregs), tumor-associated macrophages (TAMs) and MDSC [1].

## 2. Treg Cells

Treg cells, defined as CD4^+^CD25^+^Foxp3^+^ T cells, are a subpopulation of lymphocytes that are crucial in maintaining tolerance to self-antigens and innocuous foreign antigens under physiological conditions, but can also be co-opted by tumor cells to avoid the host immune response [4]. Treg cells accumulate around precursor lesions and tumor cells, inhibit tumor-specific T cell responses and impede successful immunotherapy in most cancer types, including PCa [5]. A variety of mechanisms for Treg cell-mediated suppression of effector T cell responses have been proposed, including the direct elimination of effector T cells by granzymes and perforines, secretion of inhibitory cytokines such as interleukin (IL)-10 or transforming growth factor (TGF)-β, inhibition of CD8^+^ effector T cells through membrane-bound TGF-β and competition for access to antigen-presenting dendritic cells (DCs) [6]. Tregs also affect effector T cell function by interfering with cell metabolism via the deprivation of IL2 and the promotion of adenosine production in the tumor microenvironment [6].

The prognostic significance of Tregs in cancer remains controversial in some tumor types. High levels of intratumoral Treg infiltration have been associated with poor overall survival rates in PCa and the majority of solid malignancies [7]. In contrast, increased levels of intratumoral Treg cells in colorectal, head and neck and esophageal cancer were associated with improved disease prognosis and higher overall survival [7]. Further, the prognostic value of Treg cells is not only influenced by the type of cancer, molecular subtype, tumor stage and applied model, in the case of preclinical studies, also determine the immunosuppressive and antitumorigenic role of Tregs in tumors [7]. Another barrier for the use of Tregs as the prognostic factor is the lack of surface markers defining this population and the need for the quantification of the expression level of Foxp3 (forkhead box protein 3) transcription factor, that reflects the immunosuppressive capacity of Treg cells.

## 3. The Immunotherapies in Cancers

Current immunotherapies enhance the functions of effector T cells via targeting PD-1/PD-L1 and/or CTLA-4 receptors [8]. However, Treg cells in the tumor microenvironment also express both PD-1 and CTLA-4 even at higher levels than effector T cells. Indeed, the PD-1 blockade significantly promoted the proliferation of immunosuppressive PD-1^+^ effector Treg cells in gastric cancer patients with hyperprogressive disease after nivolumab treatment [8,9]. In contrast, the CTLA-4 blockade with imilimumab mediates the selective depletion of Treg cells and enhanced CD8^+^ T effector cell cytotoxicity despite CTLA-4 expression by these two functionally opposing T cell subpopulations [8,10]. These findings suggest that a combination of PD-1 and CTLA-4 inhibitors is likely to synergize and activate intratumoral effector T cells by relieving effector T cells from PD-1/PD-L1-mediated anergy and depleting intratumoral Treg, respectively [8].

Another alternative strategy for enhancing PCa response to immunotherapy and revigorating the local immune system, is the depletion of immunosuppressive cell subsets. Accumulated evidence suggests that intratumoral Treg cells are a major obstacle for the efficacy of the immune checkpoint blockade in solid tumors [8]. First, the ratio of Treg cells to the total CD4^+^ T cells in the tumor is higher than that of Treg cells in peripheral blood; second, PD-1 inhibition promotes the proliferation of Treg cells in non-responsive cancer patients and lastly, the depletion of Treg cells enhances anticancer immunity and slows tumor progression in mice and humans [8]. Therefore, intensive efforts are ongoing to enhance immune checkpoint inhibition (ICI) therapies via the local depletion of intratumoral Treg cells, in order to avoid activating systemic autoimmune responses [8].

In an orthotopic implantation model, in which primary *Kras^G12D^*-expressing pancreatic ductal epithelial cells were injected into the pancreata of syngeneic *Foxp3*^DTR^ mice, the depletion of Foxp3^+^ cells upon diphteria toxin injection resulted in a marked reduction in tumor volume and prolonged overall survival [3]. Increasing numbers of translational immunotherapy approaches have been undertaken to deplete or disable Treg cells in solid tumors. These comprise targeting surface molecules that are expressed at a higher level in intratumoral Tregs than in circulating Treg cells. Among them, the activation of ICOS, 4-1BB and GITR was shown to impede Treg suppressive capacity and stimulated the cytotoxic activity of effector T cells [8]. Targetable chemokines and chemokine receptors mediating Treg cell recruitment into PCa have also been approached for enhancing ICI therapies in preclinical trials. Treg cells expressing CCR4 were shown to be attracted to CCL22 released by myeloid cells in ovarian cancer, and the anti-CCR4 mAb Mogamulizumab effectively elicited anticancer responses via the depletion of effector Treg cells in solid tumors [8]. Another experimental approach to disable intratumoral Treg-mediated immunosuppression is to convert them into effector T cells by targeting the histone methyltransferase EZH2 and/or Helios [8]. Recent studies on PCa have shown that, contrary to previous studies of the peripheral immune profile of unresectable PCa, a patient with a high density of Foxp3^+^CD4^+^ Tregs before the administration of neoTx showed prolonged overall survival [11]. A recent study also showed that a specific subpopulation of Treg cells with a CD45RA^−^Foxp3^low^ phenotype also correlated with better clinical outcomes in colorectal cancer patients.

In conclusion, to target Tregs in the pancreatic microenvironment, a deeper phenotyping of Treg subpopulations, including the quantification of Foxp3 expression levels throughout tumor progression is needed for the optimization of therapeutic strategies aiming to deplete or disable intratumoral Tregs [11]. Depletion of the highly immunosuppressive intratumoral Foxp3^hi^ Treg cells, could thus be used as an effective anticancer therapy, whereas strategies that locally increase the Foxp3^lo^ non-Treg subpopulation could be used to prevent tumor progression [12].

Treg cells closely interact with a subset of immature immune cells termed MDSC forming a mutually activating functional crosstalk [13]. These cells possess a highly immune suppressive machinery and are able to dampen both innate and adaptive immune responses. For instance, MDSCs inhibit the tumoricidal activity of effector T cells leading to the failure of efficient anti-tumor responses [1]. Soluble factors produced by both MDSCs and Tregs form positive feedback loops that promote the expansion of each population, boosting the suppressive phenotype of the pancreatic tumor environment [6]. In a mouse colon carcinoma model, IFN-γ-activated MDSC were shown to promote the de novo development, expansion and recruitment of Treg cells which could be explained by the up-regulation of MHC-II, IL-10 and TGF-β [1,6]. In addition, the expression of surface molecules by MDSCs including CD40/CD40L, PD-1/PD-L, and CD80/CTLA-4, promote the accumulation of Tregs and is required to induce T-cell tolerance. In a mouse ovarian cancer model, MDSC enhanced the expression of CD80 which could bind to CTLA-4 on Tregs to reinforce the immunosuppressive phenotype [6]. On the other hand, Tregs are also modulators of MDSCs expansion and protumorigenic function. Tregs enhanced the proliferation of MDSC through a TGF-β-dependent mechanism. Additionally, IL-35-producing Tregs promote the MDSC-suppressive functions via the PD-L1 pathway [6]. Furthermore, in vivo depletion of MDSC in an orthotopically transplanted model of PCa led to a concomitant decrease in Treg infiltration in the tumor [1].

MDSC can be subdivided into two major subsets, monocytic MDSC (M-MDSC) and polymorphonuclear (PMN) MDSC, which morphologically and phenotypically resemble monocytes and neutrophils, respectively [2]. Increased frequencies of both M- and PMN-MDSC are associated with a poor prognosis and metastatic dissemination in most cancer types [14]. Over the past decade, MDSC-target therapy has been explored and proven as a promising strategy for enhancing anti-tumor immunity [13]. Early work in mouse models targeted MDSCs using zoledronic acid, which reduces MDSCs recruitment through inhibition of matrix metalloproteinase 9 (MMP9) [2]. Administration of zoledronic acid resulted in prolonged survival, delayed tumor growth and increased infiltration with CD8^+^ T cells in a murine model of PCa [2]. Furthermore, the inhibition of CXCR2, a receptor found on MDSC that regulates MDSC recruitment, in a genetically engineered mouse model of PCa also resulted in enhanced survival and increased tumor infiltration with cytotoxic T cells [2]. mAb-mediated depletion of the PMN-MDSC subset with anti-Ly-6G resulted in tumor cell death and increased CD8^+^ T-cell infiltration. These findings demonstrate that MDSC-targeted therapies can partially reverse immune suppression in solid tumors and may contribute to revoke therapy resistance in PCa [2].

Tumor-infiltrating Tregs and macrophages have also been shown to synergize in order to favor an immune repressive tumor milieu. TAMs’ functional and morphological features in the pancreatic tumor microenvironment are dynamically changeable with an M1 and M2 polarization that reflect a rather proinflammatory and anti-inflammatory phenotype, respectively [8]. In PCa, TAMs are inclined to M2 deviation with protumorigenic effects, such as promoting tumor progression, enhancing immunosuppression, accelerating metastasis and inducing resistance to chemotherapeutic drugs. In the initial phase, macrophages can implement their innate immune functions to eliminate tumor cells, while this antitumorigenic role will be reversed by tumor cells throughout tumorigenesis [8,15]. To this regard, cancer cells express the “do not eat me” CD47-SIRPα molecule on their surface which leads to the impaired phagocytosis of macrophages. Pancreatic cancer cells can also impair the production of tumoricidal factors such as TNF-α and NO [15]. M2-polarized macrophages are characterized by the expression of immune suppressive cytokines such as IL-10, TGF-β, IL-6, PGE, CCL2, CCL17 and CCL20, which inhibit CD8^+^ T cell-mediated antitumor immunity and promote the differentiation and maturation of Treg cells from CD4^+^ T lymphocytes [8,15]. Given the multi-faceted role of TAMs in promoting PCa progression and their correlation with a worsened prognosis, macrophages constitute an attractive target to improve antitumor immunity and even clinical therapy [15]. Indeed, TAM-targeting therapies have seemed promising in preclinical studies, and some of these agents are currently under clinical evaluation. Therapeutic strategies targeting TAMs in PCa include macrophage depletion by blocking CSF1R signaling, inhibition of the recruitment of macrophages into the tumor microenvironment via blocking CCL2/CCR2 signaling and macrophage reprogramming towards tumoricidal classically activated phenotype via CD40 agonists or inhibition of the “do not eat me” CD47-SIRPα signaling axis to promote tumor cell phagocytosis [15].

Lastly, Tregs demonstrate a functional interplay with tumor-associated CD11c^+^ DCs and impair effector T cell function by reducing their expression of T-cell activating surface molecules. Treg cell depletion led to the restoration of immunogenic tumor-associated DCs and increased CD8^+^ T cell activation in a murine model of PCa [5].

It has become clear that Tregs and other immune suppressive cells influence tumor growth through a number of different molecular mechanisms that act either directly on cancer cells or via the inhibition of effector immune cells, most notably CD8+ cytotoxic T cells [4]. These cellular mechanisms form a partially redundant network that confers robustness to tumor immunosuppression and a compensatory network that contributes to resistance to mono-immunotherapy approaches [4]. Therefore, preclinical efforts to deplete immunosuppressive subsets in the tumor microenvironment, while initially beneficial, often result in a compensatory boost of intratumoral infiltration of other immunosuppressive cell subsets. In a recent study on a mouse model of colorectal cancer, genetic ablation or pharmacological blockade of colony-stimulating factor 1 receptor (CSF1R)^+^ macrophages resulted in an increase in tumor-infiltrating Treg cells, limiting the antitumor activity of cytotoxic CD8^+^ T cell [4]. Reversely, the depletion of Treg cells using phosphoinositide 3-kinase δ (PIK3δ) inhibitors resulted in a significant increase in CSF1R^+^ tumor-associated macrophages (TAM), which again led to the suppression of CD8^+^ T cell function [4]. Importantly, the genetic inactivation of PI3Kδ in Treg cells mediated a sensitization of tumor cells to the depletion of CSF1R^+^ TAMs and combinatorial approaches that simultaneously inhibited CSF1R and PI3Kδ substantially synergized to impede tumor progression [4]. Furthermore, in a Kras-based transgenic murine model of late-stage PCa (KPC; *FoxP3^DTR^* mice), where Tregs can be depleted at will upon the administration of diphteria toxin in the context of spontaneous carcinogenesis, Treg depletion led to a compensatory recruitment of MDSC and other CD4^+^ T cells. The expansion of MDSC within the tumor microenvironment after Treg depletion occurred upon the reprogramming of cancer-associated fibroblasts (CAFs) which promoted local immunosuppression and accelerated tumor progression [3]. The role of CAFs in PCa has been the subject of controversy over many years, the description of functionally distinct, heterogeneous CAF subsets has provided an explanation for the contradictory results obtained after the depletion of fibroblasts. In contrast to the results obtained after the depletion of FAP^+^ CAFs on a murine model of PCa, which presented delayed tumor growth and improved response to immunotherapy, the depletion the αSMA^hi^ myCAF subset resulted in increased tumor progression and decreased survival [16]. The αSMA^high^ myCAF population has been described as tumor-restraining and is predominantly driven by TGF-β signalling [3]. Since Tregs are a key source of TGF-β ligands, Treg depletion mediates a reprogramming of pancreatic CAFs from αSMA^high^ tumor-restricting myCAFs to a tumor-promoting fate [3]. Reprogrammed fibroblasts showed an increased secretion of chemokines that act as a chemoattractant for MDSC [3]. The sustained immunosuppression was countered via CCR1-inhibition, a receptor for MDSC, which further indicates that myeloid cells promote pancreatic tumorginesis and have complex and compensatory roles in the pancreatic tumor microenvironment [2]. These findings highlight the potential for therapeutic approaches targeting various immune suppressive subsets simultaneously, to each overcome the individual blockade of the other. In order to accurately target the appropriate Treg subpopulation and to design a new form of combinatorial immunotherapy able to circumvent compensatory immunosuppressive networks, both the applied mouse model (spontaneous vs. transplanted tumor) and the stage of the disease (early vs. late-stage tumors) should be taken into consideration [4].

Conventional chemotherapy has been also shown to play an immunomodulatory role on a variety of solid malignancies, including PCa. In cervical and colorectal cancer, platin-based chemotherapy led to a selective decrease in Foxp3^+^ T cells without compromising CD8^+^ T cell cytotoxicity [17,18]. Treatment with gemcitabine led to a decreased frequency of circulating M- and PMN-MDSCs in human PCa patients; however, this reduction was reversed after a resting phase without the application of gemcitabine. This demonstrates that the continuous administration of gemcitabine is needed to achieve a durable effect on the intratumoral infiltration with MDSC. This is inexorably linked to increased chemotherapy-derived side effects and a reduction in the patients’ quality of life [19]. A deeper understanding of the complex cellular crosstalk leading to Treg- and MDSC-mediated immunosuppression within the pancreatic tumor microenvironment could lead to the design of combinatorial therapies with less adverse effects than conventional chemotherapy.

One interesting and relevant context may be the tumors of patients who underwent neoadjuvant chemo-/radiotherapy. Indeed, these patients typically have locally advanced cancer, and in case of response to neoTx, they often become candidates for surgical resection. Hence, in the case of neoadjuvantly treated PCa patients, one has the unique opportunity to study the immune cell responses prior to and after neoTx within the tumor samples of the same patient. In a recent study, we showed that PCa patients who responded to neoTx and were eventually surgically resected, presented a typical immunoediting pattern compared to patients who had an upfront resection, i.e., without any prior therapy [20]. Current first-line therapies for PCa are the chemotherapy regimens modified (m)FOLFIRINOX or gemcitabine/nab-paclitaxel, which provide a modest survival benefit [2]. We demonstrated that, irrespective of the applied neoadjuvant regimen, chemotherapy led to decreased numbers of intratumoral Tregs and MDSC populations, which were accompanied by reduced aSMA^+^ myCAFs, and by increased proportions of CD8^+^ cytotoxic T lymphocytes in the tumor. Furthermore, there was also an increase in the amount of CD103^+^ DCs, which have been proposed as crucial actors for the promotion of the CD8^+^ T-cell effector activity (Figure 1) [5]. TAMs seemed to play a minor role in the immunologic reactivation mediated by neoTx on PCa patients with locally advanced tumors, as the density of both M1- and M2-polarized macrophages remained unaltered after therapy [20]. In mice, Treg ablation led to increased CD8^+^ T cell activation by harnessing the immune stimulatory potential of tumor-associated DCs [5]. Interestingly, there was no difference in the amount of CD4^+^ T cells in the tumors of primary resected versus neoadjuvantly treated patients. In fact, a higher tumor infiltration degree by CD4^+^ T cells was associated with a favourable prognosis and a survival benefit in PCa patients [20].

Even in locally advanced tumors, perioperative chemotherapy led to a parallel decrease in both Treg and MDSC which was associated with tumor shrinkage and prolonged survival. The results obtained in preclinical studies where after Treg depletion, MDSCs concomitantly decreased in early tumors whereas the inverse association was seen in advanced PCa, urge a comprehensive analysis of the phenotype and frequency of these major immunosuppressive cell subsets through pancreatic tumorigenesis. Such a “human” example demonstrates that the reaction of Treg cells and MDSC in locally advanced PCa may not actually be antagonistic but concurrent, and that these two immune cell populations should not be perceived as being mutually suppressive or substituting. The question whether the compensatory network between Treg cells and MDSCs intensifies in patients with late-stage metastatic PCa, remains yet unexplored.

Therefore, the underlying mechanisms behind the complex interplay between myeloid cells and Treg cells still needs further elucidation (Figure 2). A better understanding of the pathways behind Treg- and MDSC-mediated immune suppression may uncover novel targetable molecules that, in combination with ICI therapy, can potentially overcome the pathognomonic pancreatic immune suppression and therapy resistance [2]. To understand these dynamics, we believe that pre-clinical models that allow the administration of neoTx to different stages of PCa may be a very useful and viable option [21]. Moreover, such models may also help uncover clinically actional strategies involving Tregs for upcoming stage-specific immunotherapy trials of PCa.

## Figures and Tables

**Figure 1 cancers-14-04582-f001:**
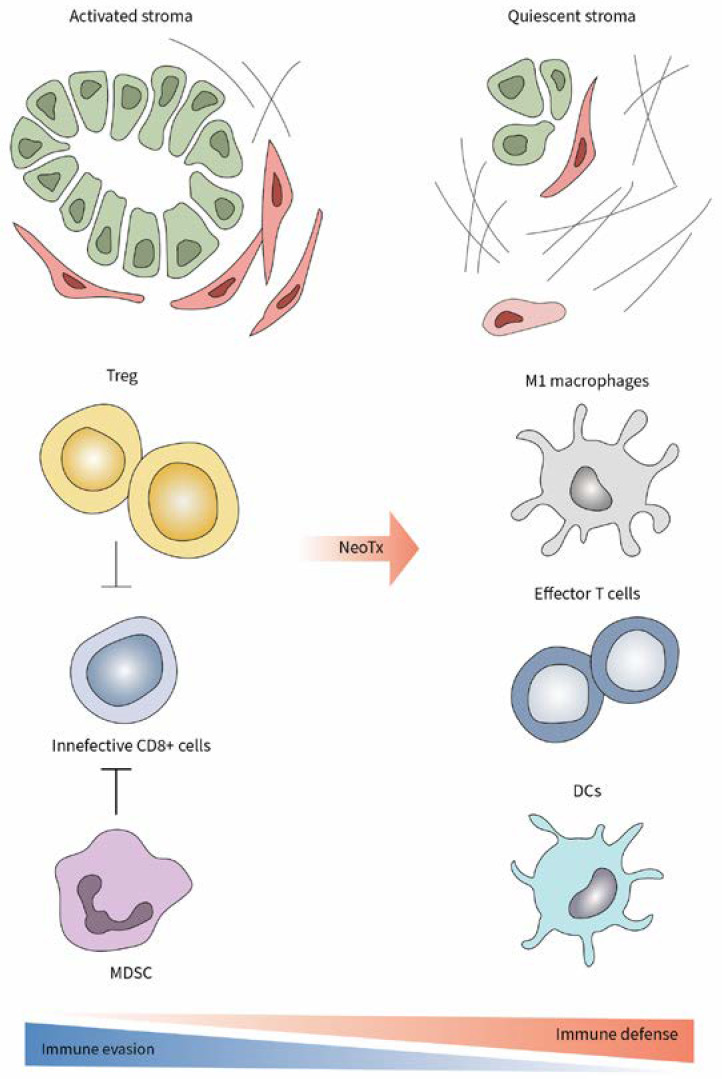
Enhancement of antitumor immune response and decreased stromal activation after neoTx in PCa.

**Figure 2 cancers-14-04582-f002:**
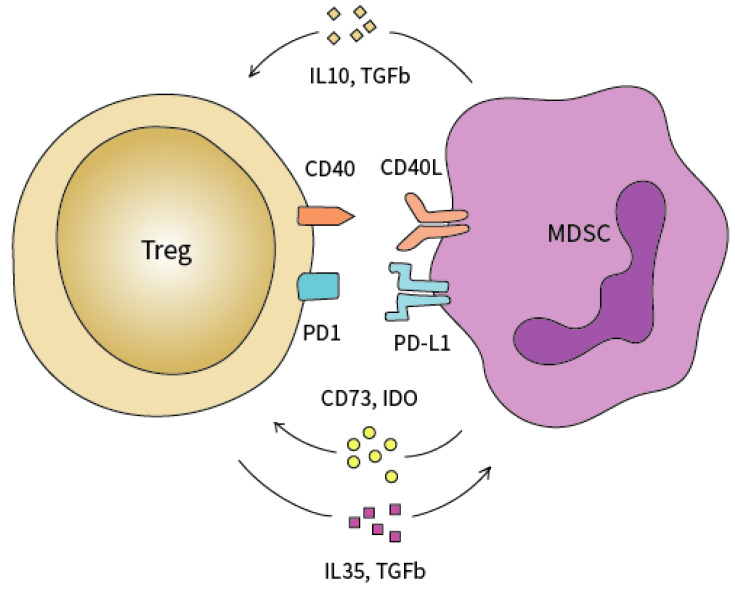
Functional crosstalk between Tregs and MDSC. MDSCs attract Tregs into the tumor microenvironment via TGF-β, IL10, CD73, and IDO secretion. Tregs modulate the expansion of MDSCs through the secretion of IL-35 and TGF-β.

## Data Availability

Not applicable for this manuscript.

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
