# Peer review of "Regulatory T Cells in Pancreatic Cancer: Of Mice and Men"

_cancers, 2022, doi:10.3390/cancers14194582_

Round 1

Reviewer 1 Report

The article by Mota-Reyes et al. describes how immune cell interactions in cancers establish an immune-suppressive environment. Also described are how immune-modulating treatments impact tumor immune responses, how these treatments can fall short, and possible approaches to improve current therapies. Overall, the writing is good with numerous examples of experimental results cited. At points the text becomes dense: the readability of the article would benefit from breaking the paper into sections. Finally, a few sentences on how the neoTx treatment works would be helpful.

Author Response

We thank the Reviewer for this important remarks. We have accordingly divided the revised manuscript into seven (7) sections. Furthermore, we have added the following sentences on the mechanism of action of neoTx on page 6, paragraph 1, of the revised manuscript:

“Classically, neoTx is believed to downsize tumors over a direct cytotoxic effect on cancer cells, since neoTx is still performed as administration of directly cytotoxic drugs such as 5-fluorouracil or platin derivatives. However, the impact of neoTx on the components of the tumor microenvironment have not yet been investigated and, as such, not unveiled (20).”

Reviewer 2 Report

The communication paper describes the regulatory T cells and interactions with myeloid suppressor cells in pancreatic tumor microenvironment in patient and mouse model. And provides critical insights into the preclinical therapy.  Overall it’s a good paper. Here is my comments:

Could authors discuss more about how Treg depletion in affects tumor microenvironment and tumor progression?

A graphical abstract is recommended to show the Treg in mouse and patient therapeutic models.

Line 31, Use the full name of pancreatic cancer instead of PCa when first introduce concept.

Author Response

We thank the Reviewer for these important comments. Accordingly:

  • The revised manuscript contains the following section on the mechanism of how Treg depletion affects tumor microenvironment and tumor progression:

“Further mechanisms of tumor progression after Treg depletion: a role of can-cer-associated fibroblasts (CAFs)

In a Kras-based transgenic murine model of late-stage PCa (KPC;FoxP3DTR mice),  where Tregs can be depleted at will upon the administration of diphteria toxin in the context of spontaneous carcinogenesis, Treg depletion led to a compensatory recruit-ment of MDSC and other CD4+ T cells. In fact, the expansion of MDSC within the tu-mor microenvironment after Treg depletion occurred upon reprogramming of cancer-associated fibroblasts (CAFs) which promoted local immunosuppression and ac-celerated tumour progression (3). The role of CAFs in PCa has been subject of contro-versy over many years, the description of functionally distinct, heterogeneous CAF subsets has provided an explanation for the contradictory results obtained after the depletion of fibroblasts. In contrast to the results obtained after the depletion of FAP+ CAFs on a murine model of PCa, which presented delayed tumor growth and im-proved response to immunotherapy, the depletion the αSMAhi myCAF subset resulted in increased tumor progression and decreased survival (16). The αSMAhigh myCAF population has been described as tumor restraining and is predominantly driven by TGF-β signalling (3). Since Tregs are a key source of TGF-β ligands, Treg depletion mediates a reprogramming of pancreatic CAFs from αSMAhigh tumor-restricting myCAFs to a tumor-promoting fate (3). Reprogrammed fibroblasts showed an increased secretion of chemokines that act as a chemoattractant for MDSC (3). The sustained immunosuppression was countered via CCR1-inhibition, a receptor for MDSC, which further indicates that myeloid cells promote pancreatic tumorigenesis and have complex and compensatory roles in the pancreatic tumor microenvironment (2). These findings highlight the potential for therapeutic approaches targeting various immune suppressive subsets simultaneously, to each overcome the individual blockade of the other.”

  • A novel graphical abstract has already been generated and uploaded together with the revised manuscript, as attached.
  • In the corresponding line, the full term “pancreatic cancer” instead of PCa has been written.

Reviewer 3 Report

The authors comprehensively review the role of T-reg lymphycytes, their relationship with the myeloid-derived suppressor cells, as important contributors in the pancreatic microenvironment. Since the pancreatic cancer is known as a rather therapy-resistant malignancy, the stromal elements might be novel antineoplastic targets. The manuscript is correctly written, it contains all relevant information, the References are up-to-date. 

Author Response

We thank the Reviewer for these favorable comments and for appreciating the quality of our work.
